# Molecular Aspects of the Development and Function of Auditory Neurons

**DOI:** 10.3390/ijms22010131

**Published:** 2020-12-24

**Authors:** Gabriela Pavlinkova

**Affiliations:** BIOCEV, Institute of Biotechnology of the Czech Academy of Sciences, 25250 Vestec, Czech Republic; gpavlinkova@ibt.cas.cz

**Keywords:** cochlea, single-cell RNAseq, transcription factor, auditory pathways, genetic mutations

## Abstract

This review provides an up-to-date source of information on the primary auditory neurons or spiral ganglion neurons in the cochlea. These neurons transmit auditory information in the form of electric signals from sensory hair cells to the first auditory nuclei of the brain stem, the cochlear nuclei. Congenital and acquired neurosensory hearing loss affects millions of people worldwide. An increasing body of evidence suggest that the primary auditory neurons degenerate due to noise exposure and aging more readily than sensory cells, and thus, auditory neurons are a primary target for regenerative therapy. A better understanding of the development and function of these neurons is the ultimate goal for long-term maintenance, regeneration, and stem cell replacement therapy. In this review, we provide an overview of the key molecular factors responsible for the function and neurogenesis of the primary auditory neurons, as well as a brief introduction to stem cell research focused on the replacement and generation of auditory neurons.

## 1. Introduction

According to the World Health Organization, hearing impairment, the partial or total inability to hear sounds, is among the top 10 disabilities of today’s society and affects approximately 6% of the world’s population. Neurosensory hearing loss is permanent and results from the death of neurons or sensory cells that have no ability to regenerate. Auditory neurons are a critical component of the auditory pathway, as they transmit auditory information from sensory hair cells to the cochlear nucleus in the brainstem. The current effective therapies for hearing impairment utilize either hearing aids to increase hair cell stimulation or cochlear implants as a substitute for hair cells. These medical devices require the presence of functional auditory neurons in the inner ear. Therefore, recent studies focus on possibilities for neuronal replacement, including exogenous stem cell transplantation and endogenous cell source replacement. Transcriptional networks are key in controlling the regeneration or replacement of auditory neurons from stem cells. Understanding and identifying individual transcription factors involved in the development and survival of auditory neurons are crucial for future more effective treatments for hearing loss [1]. This review focuses on the latest advances in the research on auditory neurons and new perspectives on future directions of this research, particularly on molecular factors important for the development and maintenance of auditory neurons. 

## 2. Functional Diversity of Auditory Neurons 

Two distinguishable systems of neurons exist within the mammalian inner ear, known as the descending neuronal pathway and ascending neuronal pathway.

The pathway descending from the cortex is called the efferent system or the olivocochlear system [2]. While being unique to the auditory region, it consists of efferent neurons subdivided into medial olivocochlear efferents and lateral olivocochlear efferents. The efferents are derived from facial branchial motor neurons. The groups differ in the neuron bodies’ locations and degree of myelination. Thicker myelinated medial efferents form synapses with outer hair cells of the organ of Corti in the cochlea (Figure 1A). Thin, unmyelinated lateral efferents innervate the dendrites of nerve fibers connecting inner hair cells [2]. The pathway uses acetylcholine as the major neurotransmitter [3,4]. The efferent system is involved in the improvement of signal detection, the functioning of the outer hair cells, and protection of the cochlea from acoustic damage. The difference in the olivocochlear system efficiency is an essential factor of the vulnerability to permanent acoustic injury [5]. 

Neurons forming the ascending neuronal pathway innervate the sensory epithelia of the organ of Corti in the cochlea and transmit auditory information in the form of electrical signals to the brain (Figure 1A). The somata of auditory neurons form the spiral ganglion that twists within a coiled cochlear duct. Peripheral neuronal processes innervating hair cells of the organ of Corti are referred to as the dendrites, whereas the central processes are the axons (Figure 1B,C). Most of them are myelinated, which is considered to be a characteristic of the axons. The axons of the afferent neurons merge into the vestibulocochlear nerve, also known as the VIII cranial nerve, relaying acoustic information further to the central nervous system. Two types of auditory neurons have been described in the cochlea (Figure 1A; [7]). Large, myelinated, bipolar type I neurons connect the inner hair cells in the organ of Corti to the cochlear nuclei in the brain. Smaller, pseudounipolar type II neurons connect the outer hair cells to the cochlear nuclei, and their peripheral processes remain unmyelinated and thin. Type I neurons represent the majority of all neurons in the spiral ganglion (approximately 95%), and a total of 5–30 type I neurons innervate one inner hair cell in the cochlea [8]. In contrast, type II spiral ganglion neurons receive input from dozens of outer hair cells, as well as supporting cells [9,10]. Type II auditory neurons appear to play a role in pain signaling and in damage perception [11,12]. Type I neurons can be further divided into three genetically distinct subtypes, known as type Ia, Ib, and Ic [13,14,15]. The subclasses exhibit significant variations, such as differences in the expression of transcription factors, neurotransmitter receptors, channel subunits, or cell adhesion molecules. Type Ia, Ib, and Ic neurons show different spatial arrangements, as well as high selectivity for a limited range of frequencies. Additionally, type I spiral ganglion neurons exhibit differences in spontaneous firing rates (SRs). Thus, they can be classified as low-SR, medium-SR, and high-SR fibers and single inner hair cells appear to be innervated by fibers with different SRs [16,17,18] (Figure 1D). Although the total number of auditory neurons decreases with age, type Ic neurons seem to be particularly vulnerable to noise or aging [13].

## 3. Tonotopic Organization of Auditory Neurons in the Cochlea

In the auditory system, cochlear sensory hair cells are connected to the brain by spiral ganglion neurons that are organized within the cochlea in an orderly fashion according to frequency, a so-called tonotopic organization, with high frequencies at the base and low frequencies at the apex [19,20]. The position of spiral ganglion neurons along the tonotopic axis of the cochlea correlates with the input frequency received from inner hair cells [19] (Figure 2A). The phenotype of auditory neurons and, thus, tonotopic map formation are also influenced by the basal–apical neurotrophin gradient, brain-derived neurotrophic factor (BDNF), and neurotrophin 3 (NT-3) [21]. In the cochlea, both the neurotrophins BDNF and NT-3 are produced in the developing sensory epithelia, whereas their respective receptors neurotrophic tyrosine kinase, receptor, type 2 (TrkB) and neurotrophic tyrosine kinase, receptor, type 3 (TrkC) are expressed on auditory neurons. NT-3 and BDNF are expressed in opposing apical–basal gradients, with the highest levels of NT-3 in the base [21]. Neurotrophins not only affect the survival but, also, the axon guidance and maturation of the firing properties of auditory neurons [22]. 

Due to the specialized innervation within the cochlea, auditory processes respond to a narrow range of frequencies, with the maximal response at one particular frequency. It is defined as the sound frequency at which an individual auditory nerve fiber is most sensitive, the characteristic frequency. The detection of complex sounds depends on the properties of auditory neuronal subtypes, particularly on spontaneous the firing rate, as well as on the proportions of these neuronal subtypes along the tonotopic axis. The distribution of these subtypes varies based on species [23]. The tonotopic (or cochleotopic) organization is maintained throughout the auditory pathways up to the cortex [24]. The formation of a tonotopic map requires the precise projection of the spiral ganglion neuron afferents of the cochlea onto the first auditory nuclei of the hindbrain, the cochlear nuclei (Figure 2B). The tonotopy of the cochlear nuclei is imprinted via the precise distribution of auditory axons with respect to their positions of origin in the spiral ganglion [25]. 

## 4. Molecular Diversity of Auditory Neurons in the Cochlea

Recently, the transcriptomic profiling of individual cells has emerged as a powerful way to investigate cellular diversity. Using the transcriptome analysis of matured auditory neurons in the cochlea, the molecular differences between type I and type II neurons and three subtypes of type I spiral ganglion neurons were characterized [13,14,15]. Unique and combinatorial molecular profiles discriminate four distinct types of matured spiral ganglion neurons correspondingly to the classification of auditory neurons based on their functional properties. For example, differences in the expression of the *Ngfr* gene encoding the p75 neurotrophin receptor between type I and type II spiral ganglion neurons correlate with their differences in the response to neurotrophins and to injury [14]. Additionally, differential expression patterns indicate tonotopic heterogeneity within the type II neurons, indicating that apical and basal type II neurons may have distinct functions [26]. Based on transcriptome profiling, type I spiral ganglion neurons are separated into three distinct subtypes: Ia, Ib, and Ic, which compose 35%, 40% and 25%, respectively, of the total population of neurons in the mouse cochlea [13]. Additionally, the proportions of these subtypes of neurons differ along the tonotopic axis of the mouse cochlea, with a larger proportion of Ia neurons and smaller proportion of Ib neurons in the cochlear base compared to the rest of the cochlea [13]. Each neuronal subtype expresses a unique set of genes encoding Ca^2+^-binding proteins and K^+^ channel and Na^+^ channel subunits that affect their response properties based on the sensitivity to sound and spontaneous firing rate. Spontaneous and sensory inputs promote the maturation and connectivity of type I and type II and type I subclasses of spiral ganglion neurons. These processes are negatively affected by mutations in genes encoding the mechanotransduction components of hair cells. For example, mutations in *Tmie* [27] and *Pcdh15* [28] interrupt the transduction and the specification of the Ia, Ib, and Ic neuron subtypes [14]. Thus, functional mechanotransduction in hair cells is essential for the expression of molecular markers that characterize different types and subtypes of auditory neurons. Nevertheless, all four types of auditory neurons exist at birth, suggesting that initial neuronal diversification in the cochlea is independent of the activity patterns and the postnatal maturation of the organ of Corti [15]. 

Potential differences in the physiological properties and molecular expression profiles of spiral ganglion neurons may also affect how these neurons transmit signals to their targets in the cochlear nucleus subdivisions in the brainstem. Single-cell transcriptome profiling showed molecular differences in the expression genes encoding axonal proteins, including glutamate receptor *Grm8*, the exocytosis regulator *Cplx2,* and netrin family gene *Ntng1*. Differences in the expression of axonal proteins may affect the synaptic properties and connectivity for different subtypes of spiral ganglion neurons [13].

## 5. Neuronal Development in the Inner Ear 

Neuronal development proceeds in parallel with the morphogenesis of the inner ear (Figure 3). All sensory organs of the inner ear and its associated sensory ganglia derive from a single embryonic source, the otic placode. The induction and morphogenesis of the inner ear from the otic placode represent highly orchestrated processes regulated by transcription factors and signaling molecules (reviewed by [29,30]). As the otic placode invaginates and forms the otocyst, neurogenesis is initiated by the expression of proneural bHLH transcription factor Neurogenin 1 (*Neurog1*), which specifies neuronal precursors [31], followed by the expression of another bHLH transcription factor, *Neurod1* [32,33]. The initial specification of the neuroblasts within the otic epithelium is followed by the delamination of neuroblasts from the anteroventral region of the otocyst as early as embryonic day nine (E9) in the mouse embryo. Soon after delamination, neuroblasts robustly express the bHLH gene *Neurod1*, proliferate, and form a cochlea-vestibular ganglion (Figure 3A,B). Neurons seem to be the first differentiated cells in the developing inner ear in all species examined; however, the inner ear structures, including the structures for hearing, vary among species [34,35]. A critical step in neurogenesis is the segregation of auditory and vestibular neurons, as the cochlea-vestibular ganglion segregates into a medial spiral ganglion and a lateral vestibular ganglion. The molecular cues regulating the specification and segregation of auditory and vestibular neurons are not fully understood. Current evidence indicates that the development of auditory and vestibular neurons is spatially and temporally segregated before or shortly after *Neurog1* expression (reviewed by [36]). All neurons after a period of proliferation undergo their final cell divisions and begin to differentiate [37]. As neurons mature, postmitotic auditory and vestibular neurons extend their processes to their peripheral targets (the organ of Corti and the five vestibular sensory epithelia) and to the central targets (the cochlear and vestibular nuclei of the brain stem). The central projections of the inner ear neurons reach the hindbrain as early as E11.5 in the mouse [38,39]. E12.5 is the earliest embryonic day to detect segregated central projections of auditory neurons to the cochlear nucleus from the vestibular nerve in the mouse [38,39]. This is a time before the peripheral projections reach their targets. Auditory neurons mature and extend their peripheral neurites, starting in the base of the cochlea, around E12.5 in the mouse [21,39,40]. Auditory neurons express two neurotrophin receptors, TrkB and TrkC, depending on their position along the axis of the cochlea, suggesting that these molecular differences in axons from different regions of the cochlea guide the topographic map formation [21]. Both receptors present in the developing auditory neurons and their respective neurotrophins (BDNF and NT-3), expressed by the sensory epithelia, are crucial not only for axon guidance but, overall, for neuronal survival, as well as synaptogenesis and the maturation of firing properties (reviewed in [22]). 

## 6. Transcriptional Network in the Development and Maintenance of Neurons in the Inner Ear

The specification, migration, differentiation, and survival of the inner ear neurons are regulated by complex networks of transcription factors and signaling molecules. The utilization of transgenic mouse models has revealed the roles of a number of essential transcription factors that govern the temporal and spatial triggers of the specification, differentiation, and maturation of the auditory neurons in the inner ear. Although two neuronal and sensory main cell fate decisions are made sequentially, they are interdependent, as regulatory networks that orchestrate the formation of the ganglia and sensory organs of the inner ear are interlinked.

### 6.1. SRY (Sex-Determining Region Y)-Box 2 (SOX2) in Neurogenic and Sensory Progenitors

SOX2 is essential for the maintenance of an undifferentiated proneurosensory cell state in various developmental systems. In the developing ear, all precursors with the ability to proliferate and differentiate express SOX2 [42]. Although SOX2 is expressed in ear placodal cells, it is not necessary for otic placode invagination and the formation of the otocyst [39]. The necessity of SOX2 in sensory development to maintain the sensory precursors has been established [43,44,45]. Although the expression of *Sox2* through the otic placode overlaps with the earliest forming neurons, its role in neurogenesis is less clear. A lack of SOX2 expression in *Lcc* mutants (light coat and circling, a mutant with an X-ray irradiation-induced mutation in an enhancer of *Sox2* in the developing inner ear [43]) has been correlated with a late absence of neurons at E15.5 [46,47]. Additional compelling evidence exists that the delayed deletion of *Sox2* results in truncated neurogenesis in the cochlea [41,42]. The delayed deletion of *Sox2* generated by *Isl1^Cre^* results in poorly differentiated Myo7a^+^ sensory hair cells only in the basal cochlear turn with severely reduced and altered innervation (*Sox2CKO*; Figure 4A,B) [41]. However, a recent study showed that the elimination of *Sox2* at the level of the otic placode has no effect on early neurogenesis, *Neurod1* expression, and the formation of the vestibular ganglion with peripheral and central projections [39]. In contrast, no neuronal innervation was found in the severely undercoiled cochlea [39], suggesting differential requirements for the presence of SOX2 in auditory neuronal development. The lack of sensory development and missing sensory epithelium-produced neurotrophic support in the absence of *Sox2* result in a massive apoptosis of all ear neurons. The interactions of SOX2 with other transcription factors likely represent a potential mechanism allowing for different roles of SOX2 in different cell lineages during ear development [42,48,49,50,51].

### 6.2. bHLH Transcription Factors in Neurogenesis

Both the bHLH factors NEUROG1 and NEUROD1 promote the neuronal fate in the inner ear. *Neurog1* is an essential gene for the induction of neurogenesis, as eliminating *Neurog1* results in a complete absence of inner ear neurons [31,53]. The absence of *Neurog1* not only eliminates neuronal development but, also, affects the sensory development, resulting in a significantly shortened cochlea with multiple rows of inner and outer hair cells in the apex [31,37]. *Neurog1*-expressing precursors within the proneurosensory domain of the otocyst also differentiate into sensory cells [54], suggesting the incomplete segregation of the sensory and neuronal fates of the *Neurog1^+^* precursors. Thus, some precursors can differentiate either into sensory cells or into neurons. 

NEUROD1 is highly expressed in delaminating neuroblasts, and *Neurod1* null mutants lose eventually most of the sensory neurons [32,55]. Like *Neurog1*-null mice, *Neurod1*-null mutants have a shorter cochlea and disorganized sensory cells in the apex [33]. The delayed deletion of *Neurod1* from developing neurons after their delamination shows that NEUROD1 participates in the segregation of vestibular and auditory neurons and in the regulation of peripheral and central projections [6]. The organization of the organ of Corti of this *Neurod1* conditional deletion mutant is comparable to the control, with three rows of outer hair cells and one row of inner hair cells (*Neurod1CKO*; Figure 4C). Although dense radial fibers are found in the mutant cochlea, noticeable abnormalities associated with the reduction and aberrant migration of neurons exist (e.g., increased spacing between radial fiber bundles, no intraganglionic spiral bundle formed by efferent axons, and disorganized central axons). Thus, the absence of *Neurod1* affects the tonotopic organization of the cochlea, resulting in a disorganized primary tonotopic map and altered features of acoustic signal processing in the central auditory system [6]. Additionally, a novel *Neurod1* and *Atoh1* double-deletion mutant demonstrates that *Neurod1*-deprived auditory neurons form projections without any sensory cells present in the cochlea after the *Atoh1* elimination (*Neurod1;Atoh1CKO*; Figure 4D) [52]. This study suggests that auditory neurons with *Neurod1* deletion are able to grow axons and form neuronal projections cell autonomously [52]. Thus, NEUROD1 has a particularly profound role in the regulation of neuronal branching, organization of projecting afferent and efferent fibers, and tonotopic organization of the auditory system.

### 6.3. ISL1 in the Development of Neuronal and Sensory Lineages in the Inner Ear

ISL1 is a Lim homeodomain transcription factor that can bind to DNA in the form of monomeric or heteromultimeric transcription factor complexes, “the LIM code” [56]. ISL1 contributes to the development of neuronal and non-neuronal cell types, including motor, sensory, and sympathetic neurons [57,58] and pancreatic endocrine [59] and cardiac cells [60]. During ear development, ISL1 is expressed in the differentiating neurons in the cochlea-vestibular ganglion and in prosensory precursors in the ear [41,52,61]. Models with the transgenic modulation of *Isl1* expression indicate the important roles of ISL1 in the maintenance and function of neurons and hair cells in the inner ear and, also, as a possible contributing factor in neurosensory degeneration [62,63,64]. The ISL1 expression pattern suggests a role in the specification and differentiation of neurons and sensory cells; however, the early embryonic lethality of loss-of-function *Isl1* mutants has prevented a full evaluation of its role in ear development.

### 6.4. Auditory Neuron Diversity Determined by Transcription Factor Combinations 

Recent single-cell RNA sequencing identified transcription factors specific for different types of auditory neurons [13,14,15]. For example, *Pou4f1*, a member of the POU (Pit-Oct-Unc) family of transcription factors, is more strongly expressed in type I neurons, with the highest levels in the Ic type [15]. Initially, *Pou4f1* is broadly expressed in all auditory neurons with the onset of neurogenesis, and its germline deletion results in significant pathfinding defects and a partial loss of neurons [65]. A subset of auditory neurons, which retain the expression of *Pou4f1* beyond the onset of hearing, play an instructive role in presynaptic properties [66]. 

In contrast, *Runx1* (runt-related transcription factor 1) is expressed mainly in Ia and Ib types of neurons, although its function is not clear [15]. In certain neural progenitor cell populations, *Runx1* controls neuronal differentiation and increases the expression of *Neurod1* [67], as well as neurotrophin sensitivity and axonal targeting [68]. 

The type II auditory neurons, representing 5% of neurons in the spiral ganglion, are morphologically and also molecularly different from type I neurons. They are not essential for the transmission of acoustic information but seem to have a role in the damage response and pain signaling [11,12]. They are postnatally characterized by specific expressions of *Etv4* and *Prox1* and increased expressions of *Mafb* and *Gata3* [13,14,15]. During development, *Gata3* is first expressed in all inner ear neurons. The early elimination of *Gata3* results in no neurosensory cell differentiation in the cochlea [69], whereas delayed conditional *Gata3* deletion severely disrupts cochlear wiring and causes disorganized and prematurely extended neurites in the cochlea [40,69]. Although *Gata3* is expressed in mature type II auditory neurons, its functional role in these neurons is still unclear.

## 7. Cell-Based Therapy for Neuronal Replacement

As sensory cells and auditory neurons have little ability to regenerate, it is only recent advances in stem cell engineering that are increasing the feasibility of cell-based therapies for hearing disorders. For future advances in the treatment of hearing disorders and cochlear implant technology, the development of methodologies for the replacement of auditory neurons represents tremendous clinical potential. This goal can be only achieved by understanding normal neuronal development, as well as the mechanisms involved in the dysfunction and degeneration of neurons in the inner ear.

One direction for the replacement of neurons is to reprogram endogenous cells in the inner ear, which would be more physiological. Proliferative cells in the inner ear showing stem cell properties were identified by several groups [70,71]. However, the population of stem cells in the cochlea is extremely limited. A different attempt was made by inducing neuronal fates in non-sensory epithelial cells in the cochlea with the virus-mediated ectopic expression of *Neurog1* and *Neurod1* [46]. Although *Neurog1* and *Neurod1*-transfected cells demonstrate neuronal phenotypes, this transcription factor-induced reprogramming is not sufficient on its own to generate mature neurons in cochlear explants [46]. Additionally, the ability to induce a neuronal fate in non-sensory epithelial cells is mainly limited to embryonic development in the mouse. Another source of new neurons might be glial cells present in the spiral ganglion. Glial cells have been reprogrammed into functional neurons using the virus-mediated ectopic expression of transcription factors and small soluble molecules in the brain [72], which represent one possible future therapy also for hearing loss.

A second direction for neuron replacement is to use exogenous pluripotent stem cells (PSCs). PSCs have the capacity to self-renew and are capable of differentiating into multiple cell types. There are two sources of PSCs: embryonic stem cells (ESCs) and induced pluripotent stem cells (iPSCs) generated by the reprogramming of somatic cells. PSCs have been used to generate inner ear cells for cell-based therapy and for in vitro analyses (Figure 5; reviewed by [73]). For cell-based therapy, PSCs might be transplanted at any stage of differentiation to avoid the prevalent apoptosis of differentiated neurons. This would also allow the introduction of local cues of the inner ear environment during neuronal differentiation for neurite growth and synaptogenesis to produce functional neurons. Several differentiation protocols to produce sensory neuron-like cells from mouse and human PSCs have been established [74,75,76,77,78]. To avoid immunocompatibility issues, using iPSCs derived from the patient’s own cells would be advantageous [79]. The current challenges to neuronal transplantation include cell survival, differentiation, and the formation of synaptic connections. Nevertheless, PSC-derived 3D human cochlear organoids represent a powerful tool to investigate the molecular mechanisms underlying hearing disorders and to examine the effects of genetic corrections of genetic mutations using, for example, clustered regularly interspaced short palindromic repeat (CRISPR)/Cas9 technology and to validate drug-based therapy (reviewed by [80]). In summary, significant progress has been made in the generation of neurons in vitro and in vivo and in transplantation protocols, yet further work is needed to optimize the restoration of hearing functions and to be able to tonotopically organize transplanted cells.

## Figures and Tables

**Figure 1 ijms-22-00131-f001:**
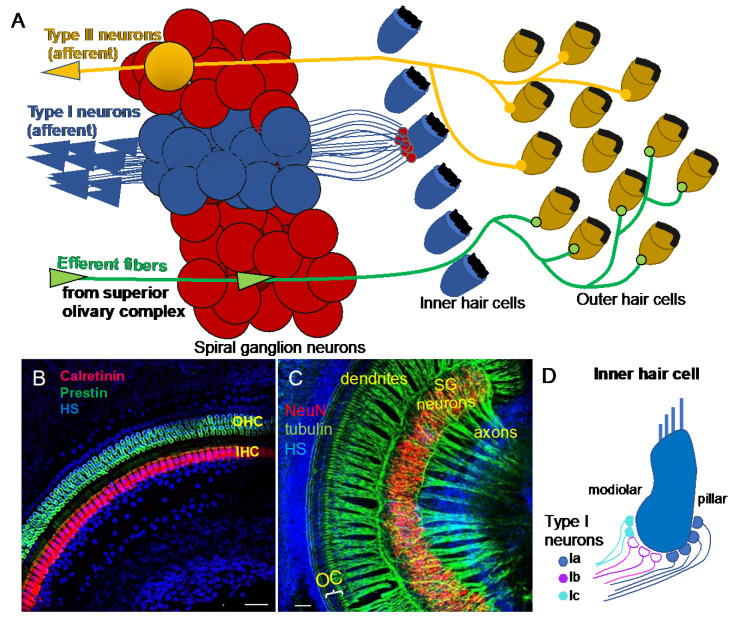
Auditory neurons form the spiral ganglion in the cochlea and connect hair cells in the organ of Corti to cochlear nuclei in the brain stem. (**A**) Diagram shows innervation of the organ of Corti. Type I neurons extend radial fibers toward the inner hair cells (5–30 type I neurons innervate one inner hair cell), and type II neurons provide diffuse innervations to the outer hair cells and supporting cells within the cochlea. Efferent axons from superior olivary complex innervate the outer hair cells. (**B**) Whole-mount immunolabeling of the mouse cochlea shows outer hair cells (OHC, labeled by anti-prestin) and inner hair cells (IHC, labeled by anti-calretinin) forming the organ of Corti (OC). (**C**) Whole-mount immunostaining of the spiral ganglion (SG) neurons (anti-NeuN, a neuronal soma marker) shows the dendrites extending in the periphery to the organ of Corti (OC) and the axons traveling to the central nervous system (anti-acetylated alpha-tubulin-labeled nerve fibers). Scale bars, 50 μm. Confocal images (**B**) taken from [6] and (**C**) unpublished data. (**D**) Schematic drawing of synapses of molecular subtypes of type I neurons on the inner hair cell. Type I neurons are segregated into three distinct groups: types Ia (blue), Ib (magenta), and Ic (green-blue) of the spiral auditory neurons, which extend peripheral processes that are spatially segregated on the surface of the IHC. HS: Hoechst nuclear staining.

**Figure 2 ijms-22-00131-f002:**
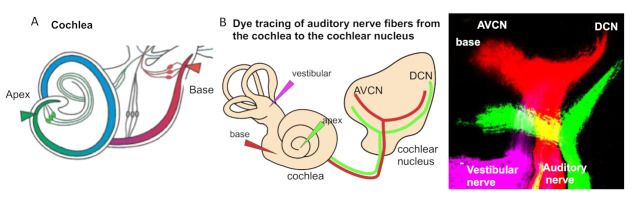
Tonotopic organization. (**A**) An illustration of the cochlea and its tonotopy across the frequency spectrum. High-frequency sounds maximally stimulate the base of the cochlea (red), whereas low-frequency sounds maximally stimulate the apex (green). (**B**) The tonotopic organization of the cochlear nucleus can be visualized by lipophilic dye tracing. Diagram of applications of different colored dyes (magenta, vestibular organs; red, base; and green, apex) to label distinct bundles of neuronal fibers of the auditory nerve projecting to the cochlear and vestibular nuclei. Representative image of triple-dye labeling from the cochlea shows the trajectory of low- (green from the apex) and high-frequency (red from the base) auditory nerve fibers and the tonotopic organization of the cochlear nucleus subdivisions, the anteroventral (AVCN) and dorsal cochlear nucleus (DCN), in mouse embryos at embryonic day 18.5. Image from Reference [6].

**Figure 3 ijms-22-00131-f003:**
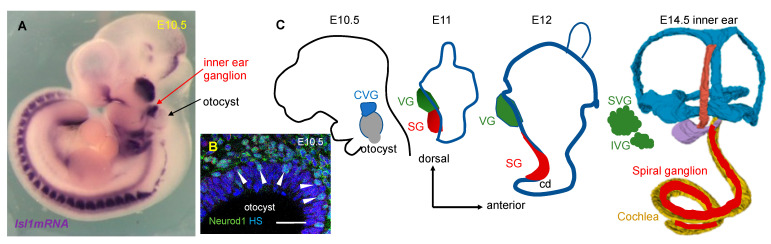
Morphogenesis of the inner ear ganglia. (**A**) Neurons forming cochleovestibular ganglion in the mouse embryo at E10.5 are visualized by the detection of *Isl1* mRNA in situ hybridization. Image from Reference [41]. (**B**) NEUROD1^+^ delaminating neuroblasts detected in the proneurosensory epithelium of the otocyst. Confocal image with permission from Reference [39]. Scale bar, 100 μm. (**C**) The stages of inner ear morphogenesis are shown schematically from the otocyst to the mature three-dimensional structure. In parallel, the neurons delaminate to form first a cochlear-vestibular ganglion (CVG), followed by the gradual separation of the vestibular (VG, green) and spiral (SG, red) ganglia, which eventually innervate the vestibular and auditory sensory epithelia. cd, cochlear duct; HS, Hoechst nuclear staining; SVG, superior vestibular ganglion; and IVG, inferior vestibular ganglion.

**Figure 4 ijms-22-00131-f004:**
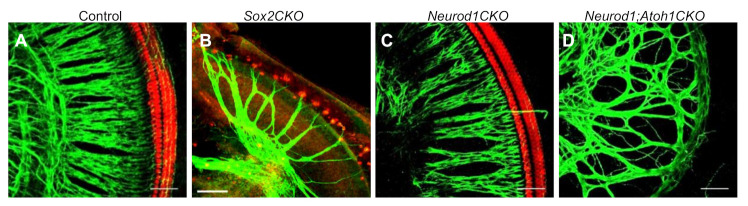
Differential innervation phenotype in the cochlea as the effect of the elimination of transcription factors. (**A**) Whole-mount immunostaining with anti-Myo7a (a marker of hair cells, HCs) and anti-ß-tubulin (nerve fibers) antibodies shows the innervation and the formation of the sensory epithelium and hair cells in the mouse control cochlea at postnatal day 0. (**B**) Delayed deletion of the SRY (sex-determining region Y)-box 2 (*Sox2*) [41], (**C**) *Neurod1* [6], and (**D**) double deletion of *Neurod1* and *Atoh1* [52] were generated by *Isl1^Cre^* and result in abnormalities in the number of neurons, formation of sensory epithelium, and cochlear innervation. Scale bars, 50 μm. (**D**) With permission from Reference [52].

**Figure 5 ijms-22-00131-f005:**
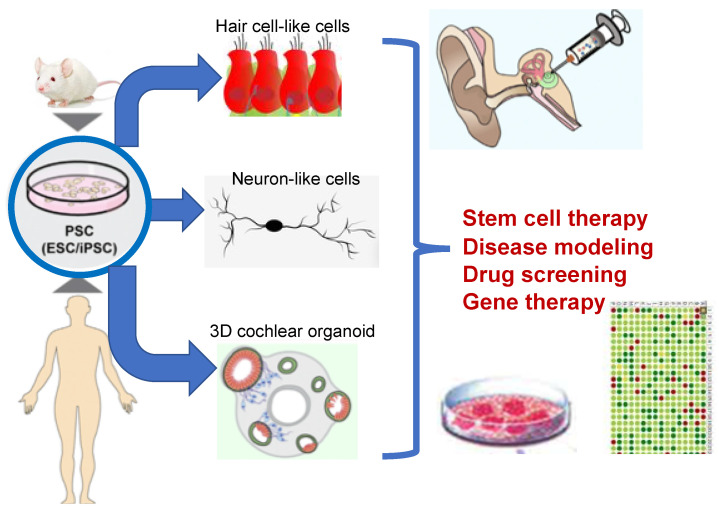
Stem cell applications. Mouse- and human-derived embryonic and induced pluripotent stem cells can be reprogramed in order to generate sensory cells and neurons for stem cell therapy and cochlear organoids for in vitro analyses (adapted from [80]). PSC: pluripotent stem cell.

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
