# Peer review of "Molecular Aspects of the Development and Function of Auditory Neurons"

_ijms, 2020, doi:10.3390/ijms22010131_

Round 1
Reviewer 1 Report
The article is a narrative informative review written in a chapter – like manner which gives an up-to date source of information on the molecular aspects and function of auditory neurons in the inner ear.
Except for the fact that it gives a lot of valuable information on the subject , I cannot see study design , study characteristics, and I cannot follow any methodology of this article as usually it is met in contemporary high value reviews or meta-analyses.
In some situation when very experienced scietist is asked by editorial board for state of art narrative review this article could be accepted. Otherwise I would seek for some form of methodology in this interesting paper.
Author Response
Dear Reviewer,
We would like to thank this reviewer for very positive comments. Indeed, this review did not include any methodology or study design.
Comments and Suggestions for Authors:
The article is a narrative informative review written in a chapter – like manner which gives an up-to date source of information on the molecular aspects and function of auditory neurons in the inner ear.
Except for the fact that it gives a lot of valuable information on the subject, I cannot see study design, study characteristics, and I cannot follow any methodology of this article as usually it is met in contemporary high value reviews or meta-analyses.
In some situation when very experienced scientist is asked by editorial board for state of art narrative review this article could be accepted. Otherwise I would seek for some form of methodology in this interesting paper.
Reviewer 2 Report
The review is concise but provides a exhaustive and meaningful overview of auditory neuron diversity, development, organization and function throughout the cochlea. It is well written and well organized, figures are clear and self explanatory. I appreciated the reading.
I have two minor suggestions:
1- auditory neuron and inner ear are redundant in the title. How about shortening the title and removing "in the inner ear"
2- I would revise the sentence line 327 of the manuscript:
"Different differentiation protocols to produced sensory neuron-like cells from mouse and human PSCs have been established"
Suggestion: several instead of Different and remove the d from produced.
Author Response
Dear Reviewer,
We very much appreciate the in depth and thorough review of our submission and would like to thank this reviewer for very positive comments.
Comments and Suggestions for Authors:
The review is concise but provides an exhaustive and meaningful overview of auditory neuron diversity, development, organization and function throughout the cochlea. It is well written and well organized, figures are clear and self-explanatory. I appreciated the reading.
1-auditory neuron and inner ear are redundant in the title. How about shortening the title and removing "in the inner ear"
- We changed the title, as suggested.
2- I would revise the sentence line 327 of the manuscript:
"Different differentiation protocols to produced sensory neuron-like cells from mouse and human PSCs have been established"
Suggestion: several instead of Different and remove the d from produced.
- Thank you. We corrected and modified the sentence (line 334).